# High power density redox-mediated *Shewanella* microbial flow fuel cells

Leyuan Zhang[1,2], Yucheng Zhang[1], Yang Liu [1], Sibo Wang [2], Calvin K. Lee[3], Yu Huang [1,4] ✉ & Xiangfeng Duan [2,4] ✉

Microbial fuel cells utilize exoelectrogenic microorganisms to directly convert organic matter into electricity, offering a compelling approach for simultaneous power generation and wastewater treatment. However, conventional microbial fuel cells typically require thick biofilms for sufficient metabolic electron production rate, which inevitably compromises mass and charge transport, posing a fundamental tradeoff that limits the achievable power density ($<1\,mW\,cm^{-2}$). Herein, we report a concept for redox-mediated microbial flow fuel cells that utilizes artificial redox mediators in a flowing medium to efficiently transfer metabolic electrons from planktonic bacteria to electrodes. This approach effectively overcomes mass and charge transport limitations, substantially reducing internal resistance. The biofilm-free microbial flow fuel cell thus breaks the inherent tradeoff in dense biofilms, resulting in a maximum current density surpassing $40\,mA\,cm^{-2}$ and a highest power density exceeding $10\,mW\,cm^{-2}$, approximately one order of magnitude higher than those of state-of-the-art microbial fuel cells.

Using exoelectrogenic microorganisms to harness the energy from biodegradable organic matter can provide an alternative source of electrical power and lay out a potentially cost-effective approach for waste water treatment[1,2]. To this end, microbial fuel cells (MFCs) that can directly convert organic matter into electricity have attracted increasing interest[3,4]. Diverse types of bacterial species have been found to produce electrical current in MFCs[5]. Among these exoelectrogenic bacteria, *Escherichia coli* DH5α, *Geobacter sulfurreducens* KN400, *Rhodopseudomonas palustris* DX1, *Shewanella oneidensis* (*S. oneidensis*) MR-1, *Shewanella putrefaciens*, and mixed culture 1CA have been reported to show the superior power output in MFCs[6], among which *Geobacter* (Gram-negative) and *Shewanella* (Gram-negative) species represent the most widely studied examples[7–10]. It is recognized that the power output from microorganisms is fundamentally limited by the intrinsic metabolic rate and the transmembrane/extracellular electron transfer processes. The transmembrane electron transport usually involves various types of cytochromes in the periplasmic space and outer membrane[11], while the extracellular electron transport (EET) from cells to electrodes may be facilitated by either the direct contact or redox mediator mediated indirect pathways[7]. Considerable efforts have been devoted to enhancing electron transfer processes, by adopting porous carbon electrodes[12–15], coating conductive polymers[16,17], fusing metallic nanoparticles[18–21] or using artificial electron mediators[22–25].

The performance of the MFCs is strongly influenced by several architectural and operational factors, including the thickness and density of the anodic biofilm, the efficient delivery of organic substrates to and extraction of metabolic electrons from individual bacteria within the biofilm, membrane conductivity, solution conditions and operating temperature[11,26–28]. Conventional MFCs typically adopt a H-type cell architecture with the anode and cathode chamber separated by an anion- or cation-conductive membrane[26]. Within this configuration, long ion transport path from the anode to cathode, together with low conductivity of bacteria medium, could cause high internal resistance and limit the total power output. The ohmic resistance in the H-type MFCs is often in the range of 100 s of ohms cm².

[1]Department of Materials Science and Engineering, University of California, Los Angeles, Los Angeles, CA, USA. [2]Department of Chemistry and Biochemistry, University of California, Los Angeles, Los Angeles, CA, USA. [3]Department of Bioengineering, University of California, Los Angeles, Los Angeles, CA, USA. [4]NanoSystems Institute, University of California, Los Angeles, Los Angeles, CA, USA. ✉e-mail: yhuang@seas.ucla.edu; xduan@chem.ucla.edu

Moreover, this design typically requires long-duration (10 s of hours) incubation of thick biofilms on a conductive electrode to maximize metabolic electron generation rate and the overall power output in a given area. The resulting biofilms are usually rather sensitive to environmental disturbance, which limits the robustness of the MFCs[29–31]. Additionally, research has indicated that prolonged incubation periods can lead to the formation of stratified biofilms, characterized by an outer layer of live bacteria and an inner layer of dead bacteria adhering to the electrode surface[32–34], due to the limited nutrient supply and/or proton accumulation near the electrode surface. Consequently, only the surface layer of biofilms inhabited by live bacteria exhibits the metabolic activity, while the inner dead layer may serve as an electrically conductive matrix. Although long-distance EET is feasible through thick biofilms, it heavily relies on the biofilm conductivity. In instances, where the electrical conductivity of biofilms (e.g., *S. oneidensis*) is restricted, the metabolic activity of the live layer may be hindered by the high charge-transfer resistance associated with long-distance EET. Hence, sustaining a high-quality biofilm with a significant proportion of metabolically active bacteria is crucial but represents a persistent challenge for achieving high-performance MFCs.

Furthermore, the need of thick biofilms in conventional MFCs to achieve adequate metabolic electron generation inherently compromises nutrient delivery and reduces electron extraction efficiency from individual bacteria within the dense biofilm, resting an intrinsic tradeoff that limits the achievable power density (Fig. 1a). Mass transport analysis predicted that substrate (nutrient) mass transport limitations in the biofilm restrict the maximum achievable power density to 1.7 mW cm$^{-2}$ in conventional MFC designs[35]. Indeed, despite considerable efforts devoted to boosting extracellular electron transport and optimizing the MFC electrode and cell architectures[36], the experimental power densities (normalized by the projected anode area) reported to date (<1 mW cm$^{-2}$) are generally below this fundamental limit[20,37], and far from conventional fuel cells that can deliver a much higher power density (2–1000 mW cm$^{-2}$). Breaking this limit necessitates not only the development of advanced engineering strategies to enhance electron transfer efficiencies but also the implementation of advanced cell architecture designs to improve mass transport and reduce internal resistance.

Herein, we report a unique design of redox mediated microbial flow fuel cells (MFFCs) by exploiting artificial redox mediators in flowing medium to efficiently extract metabolic electrons from the planktonic bacteria and deliver them to porous carbon electrodes. This design simultaneously optimizes the kinetics of electron, ion, and mass transport, greatly reducing the internal resistance and boosting the overall power output. Without the time-consuming incubation of thick biofilms, the biofilm-free MFFC breaks the fundamental tradeoff between the requirement of thick biofilms for adequate metabolic electron generation and mass/ion/electron transport limitations in dense biofilms. As a result, we realize a redox-mediated MFFC that delivers a breakthrough power density beyond 10 mW cm$^{-2}$, about one order of magnitude higher than the best value achieved in MFCs to date, which is nearly comparable to direct methanol fuel cells (2–100 mW cm$^{-2}$)[38] and some emerging energy transition technologies such as hydrovoltaic or thermoelectric/thermogalvanic power generations[39–41].

## Results
### Concept of MFFC design
A proof-of-concept MFFC configuration is schematically illustrated in Fig. 1b. It has been recognized that electrochemically active bacteria can transfer electrons to conductive electrodes via either the direct contact or indirect pathways[7]. Thus, we hypothesize it is possible to extract electrons and produce electrical current by flowing bacteria through a porous conductive electrode (e.g., carbon felt), similar to the working mechanism of typical redox flow batteries[42], in which the redox species dissolved in solutions release electrons in the anode when flowing through the electrode during the discharge process. The full-cell MFFC is constructed with two external reservoirs for the bacteria anolyte and ferricyanide catholyte (Fig. 2a), separated by an ion transport membrane sandwiched between two porous carbon felt electrodes (Supplementary Fig. 1). The *S. oneidensis* bacteria suspension with lactate as the nutrient is used as the model anolyte stored in an external reservoir, circulating between the reservoir and a porous carbon felt anode.

As the bacteria flow through the carbon felt electrode, metabolic electrons on bacteria surface are transferred to the electrode. In this way, there is no need to incubate thick biofilms on anode, and it can bypass some key challenges associated with the conventional MFCs to achieve greatly improved mass transport and charge transfer rates: (i) With the bacteria and substrate (nutrient) homogeneously dispersed in the flowing medium, there is no concentration gradient, thus eliminating mass transport limitations and ensuring efficient charge extraction from the substrate to the bacteria; (ii) By passing the bacteria through the porous carbon felt electrode, multiple collisions facilitate charge transfer from the bacteria to the electrode, improving the charge transfer efficiency without the need to traverse the highly resistive biofilm; (3) By further introducing artificial redox mediators (electron acceptors, such as flavins, quinones, viologens, phenazines, or phenothiazines)[43,44] homogeneously mixed with bacteria in the

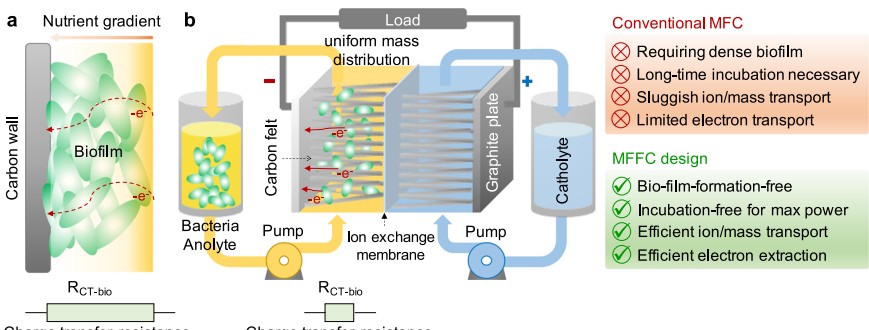

**Fig. 1 | The concept of MFFC. a** Schematic illustration of the nutrient diffusion gradient and charge transport resistance in dense biofilms. Due to mass transport limitations, substrate depletion can occur for the bacteria located near the carbon electrode, deep within the biofilm; Meanwhile, electrons extracted from bacteria on the surface of the biofilm where nutrients are abundant need to go through highly resistive biofilms. The color gradient represents the nutrient concentration gradient caused by mass diffusion limits in thick biofilms. **b** (left) Schematic illustration of a MFFC that simultaneously allows efficient nutrient delivery to the planktonic bacteria in a homogeneous medium and electron extraction from the planktonic/flowing bacteria through collision contacts with porous carbon electrodes; (right) comparison of conventional MFCs with MFFCs highlighting the potential merits of MFFCs.

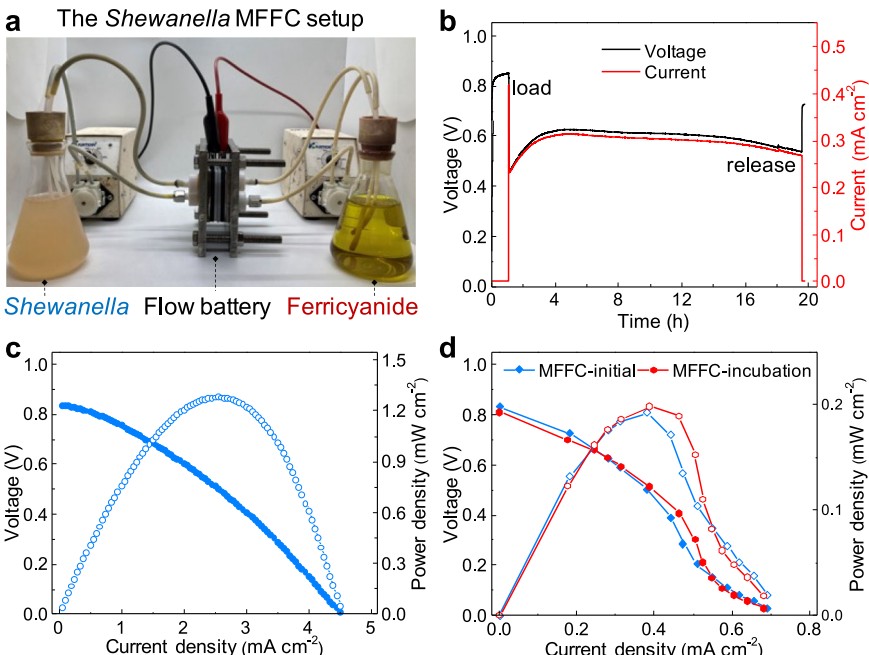

**Fig. 2 | Electrochemical characterizations of the MFFC system. a** A photograph of a *Shewanella* MFFC setup. **b** The voltage and current output from the MFFC under the open circuit and constant resistor (500 Ω) loading conditions. **c** Instant polarization curves and power output of the MFFC obtained at the scan rate of 0.5 mA cm$^{-2}$ s$^{-1}$. **d** Polarization curves and power output of the MFFC, measured with external resistors at the beginning (< 30 min) and after 16 h of operation. The external resistors applied are 995, 675, 559, 470, 330, 219, 150, 100, 68, 47, 33, 22 and 10 Ω, respectively.

medium with the number concentration 10 orders magnitude higher than that of bacteria particles (1.8 − 15 × 10$^{18}$ mediators/cm$^3$ vs. 8 × 10$^8$ bacteria/cm$^3$), the electrons produced in bacteria can be efficiently transferred to the mediators with little charge transfer limitations within the flowing medium; and it also ensures higher-frequency collision-based charge transfer from the mediators to the carbon felt electrode. In this way, we effectively bypass key challenges associated with conventional MFCs, achieving much lower mass transport resistance, greatly reduced charge transport resistance, and greatly enhanced current densities and power densities.

As the air/oxygen electrode is complicated and can be a limiting factor, a ferricyanide solution is used as the catholyte circulating between the carbon felt cathode and catholyte reservoir to simplify the system for helping analyze the anode behavior in the initial studies. The ion exchange membrane sandwiched between anode and cathode carbon felt electrodes allows ion transport between the anode and cathode chambers to balance the charge and complete the electrical circuit. The sandwich structure with ion exchange membranes between two carbon felt electrodes greatly reduces the ohmic resistance, which is critical for further boosting the overall power output. Additionally, the flow rate and flow channel may play a role on the interactions between bacteria and electrodes.

## MFFC demonstration and performance evaluation

A photograph of a proof-of-concept MFFC assembled in our study is shown in Fig. 2a. We first evaluated the voltage output from the MFFC. It is noted that the open circuit voltage (OCV) of the MFFC increases rapidly with the flowing of the planktonic bacteria (optical density, OD ∼ 0.8–1.2; flow rate ∼20 ml min$^{-1}$) and reach a nearly constant value (0.85 ± 0.05 V) within several minutes once the solution through porous carbon felt electrode and the surface wetting behavior is stabilized (Fig. 2b), indicating the effective electron transfer from the flowing bacteria to the porous electrode. According to the polarization, connecting the MFFC with a 500 Ω external resistor, the cell voltage instantly dropped to 0.47 V, which gradually recovers to 0.54–0.63 V, with a relatively stable current output of 0.3 mA cm$^{-2}$ and power

output of 0.18 mW cm$^{-2}$, suggesting a stable operation of the MFFC. Upon removing the loading resistor, the cell voltage rapidly restores to 0.7–0.8 V.

For the power evaluation, we first adopted the method commonly used in the field of redox flow batteries (Fig. 2c)[42]. At a scan rate of 0.5 mA cm$^{-2}$ s$^{-1}$, the polarization curve is obtained with a maximum current density of 4.5 mA cm$^{-2}$ and power density of 1.28 mW cm$^{-2}$, confirming the feasibility of MFFC concept. It is well known that the scan rate can affect the power result of MFCs due to mass transport limitations[45]. To further validate the robust function of the MFFC systems, we also used commonly adopted testing protocols in the MFC community by loading a series of external resistors and recording the output voltage to derive the polarization and power curves (blue line in Fig. 2d). Although the resistor-based measurements show the same OCV and a rather similar polarization behavior at low current density, the maximum achievable current density is substantially lower (0.7 mA cm$^{-2}$ vs. 4.5 mA cm$^{-2}$ in the instant measurement described above). Overall, with the resistor-based measurement, the MFFC system can show a stable maximum power density of ∼0.19 mW cm$^{-2}$ without the biofilm formation process. Microscopy analysis of carbon felt electrodes shows little bacteria attachment at the beginning (< 6 h) of operation (Supplementary Fig. 2). A 16-hour incubation led to apparent bacteria attachment and biofilm formation, which may contribute to additional power generation. Nonetheless, the power output only increased slightly to 0.20 mW cm$^{-2}$ after the incubation (red line in Fig. 2d), indicating the overall power output in the MFFC is primarily originated from the flowing bacteria rather than the immobilized biofilm, as also indicated by the mostly stable OD of the bacteria anolyte during the continuous flow (Supplementary Fig. 3).

The notable difference between the output power densities observed in the instant and resistor-based measurements under the same condition can be attributed to the kinetic limitations caused by the rapid depletion of metabolic electrons on the bacteria surface. In this case, the metabolic electrons on the bacteria surface are rapidly discharged (transferred) to the carbon felt electrode once they are in contact, and metabolic rate or transmembrane electron transfer is not

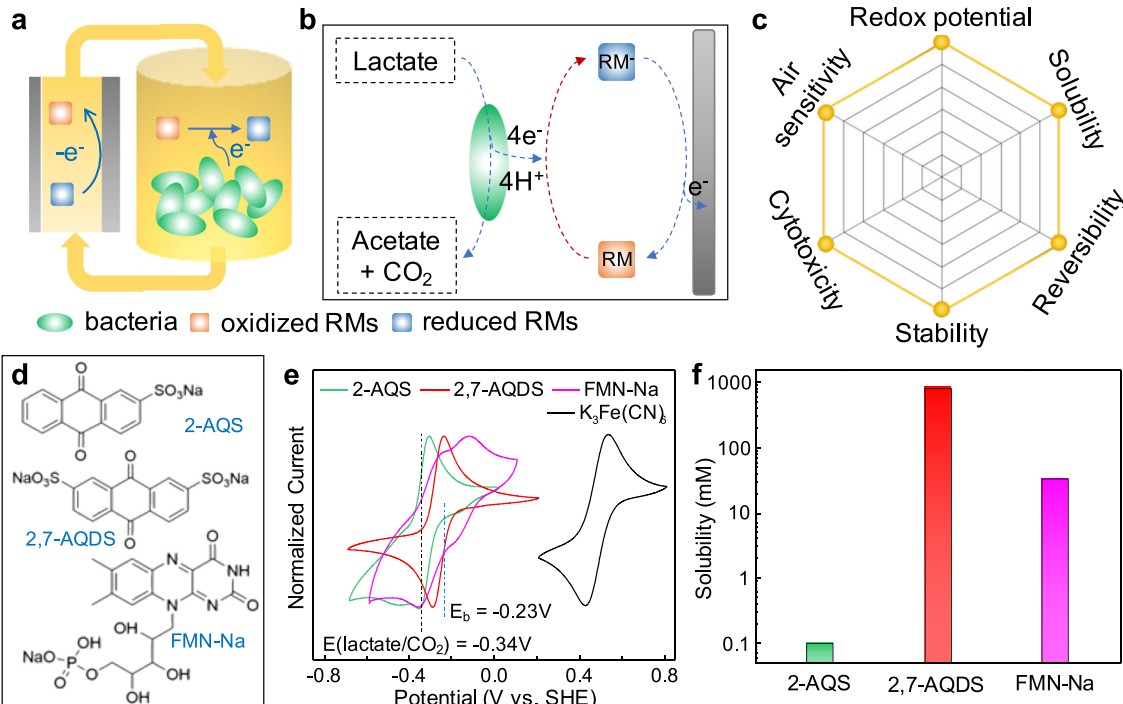

**Fig. 3 | Redox mediator strategy in MFFCs. a, b** Schematic illustration of working principle of redox mediator strategy. RM is redox mediator. **c** Important factors to be considered in screening redox mediators. **d** Molecular structures of the selected redox mediators: riboflavin 5'-monophosphate sodium (FMN-Na), anthraquinone-2,7-disulfonic acid disodium (2,7-AQDS), and anthraquinone-2-disulfonic acid disodium (2-AQS). **e** Cyclic voltammetry of different redox mediators in aqueous neutral solution in reference to the ferricyanide redox pairs. **f** Solubilities of various redox-active molecules measured in bacteria medium.

fast enough to refresh surface electrons to sustain the high constant current density under the resistor-based measurement, as also indicated by the rapidly diminished power output when the flow is interrupted in the MFFC (Supplementary Fig. 4). Nonetheless, even with the resistor-based measurement, the output power density achieved in the MFFCs (~0.20 mW cm$^{-2}$) is notably higher than that from the conventional MFCs (~0.05 mW cm$^{-2}$) constructed with the same bacteria medium mixture and carbon felt electrodes (Supplementary Fig. 5), highlighting the advantage of the MFFCs in more efficient mass transport and charge extraction. We note a similar flow setup has been used to boost nutrient delivery to biofilms in some previous MFC studies[46,47], in which cases the biofilm (instead of flowing bacteria) remains the major electron source.

**Redox mediator strategy**

With the working principle of the MFFCs, there are two possible electron transfer pathways from the flowing bacteria particles to the carbon felt electrodes. First, direct electron transfer could happen through collision contacts between the bacteria particles and the porous electrodes. In this case, the charge transfer rate and output current are dictated by the effective collision probability and ultimately by the bacteria particle concentration and flow rate. Alternatively, a redox shuttle mediated indirect electron transfer process is also widely recognized in the MFC systems, in which metabolic electrons are first transferred to redox mediators, which shuttle the electron to the conductive electrodes through redox cycles. In this case, the electron transfer rate depends on the collision interactions between the redox mediators and the bacteria as well as those between the mediators and the electrode, which is ultimately dictated by the concentration of the redox mediators[48,49].

The naturally produced redox mediators by bacteria (e.g., flavin) exhibited limited bulk concentration[23,48], and cannot function efficiently in the MFFCs. To this end, introducing additional artificial redox mediators could offer an attractive strategy to boost the overall charge transfer efficiency (Fig. 3a). The redox-active molecules in bacteria solution feature a much smaller size and much larger number density (e.g., $1.5 \times 10^{19}$ cm$^{-3}$ for 25 mM redox mediator concentration vs. $8 \times 10^8$ cm$^{-3}$ for bacteria solution at OD = 1), and can make much more efficient collision contacts with both the bacteria and the carbon felt electrodes to boost charge transfer efficiency. When the oxidized form of redox mediators come in contact with bacteria, they extract metabolic electrons from bacteria to produce the reduced form (Fig. 3b); next, when the reduced form in the flowing medium makes collision contacts with the carbon felt electrodes, they transfer electrons to the porous carbon felt electrodes and produce the oxidized form to complete the redox cycle. In this way, the redox mediators can greatly accelerate the transfer of the metabolic electrons from the bacteria to the carbon felt electrodes and boost the output current and power.

The additional redox cycle involving the artificial redox mediators would inevitably lower the output potential. There are several general guiding principles in choosing an optimum redox mediator to maximize the electron transfer efficiency while minimizing voltage losses (Fig. 3c). From the outset, to efficiently extract electrons from bacteria and simultaneously maintain high voltage output, the redox potential of the mediators should be comparable to or slightly more positive than that of bacteria. Further, a reasonable solubility is necessary to ensure sufficient concentration and maximize the collision contact probability. Additionally, other factors, including redox reaction reversibility, chemical/electrochemical stability, biocompatibility, and air sensitivity are also important factors to consider for robust operation of the MFFCs[50,51]. Though some artificial redox mediators have been previously explored, the rationale for the mediator selection is less understood and rarely optimized. Most reported mediators such as 2,6-AQDS[46], methyl viologen[50], or methylene blue[51] have different aspects of problems such as poor solubility (<1 mM), mismatched redox potentials, cytotoxicity, or extreme air sensitivity.

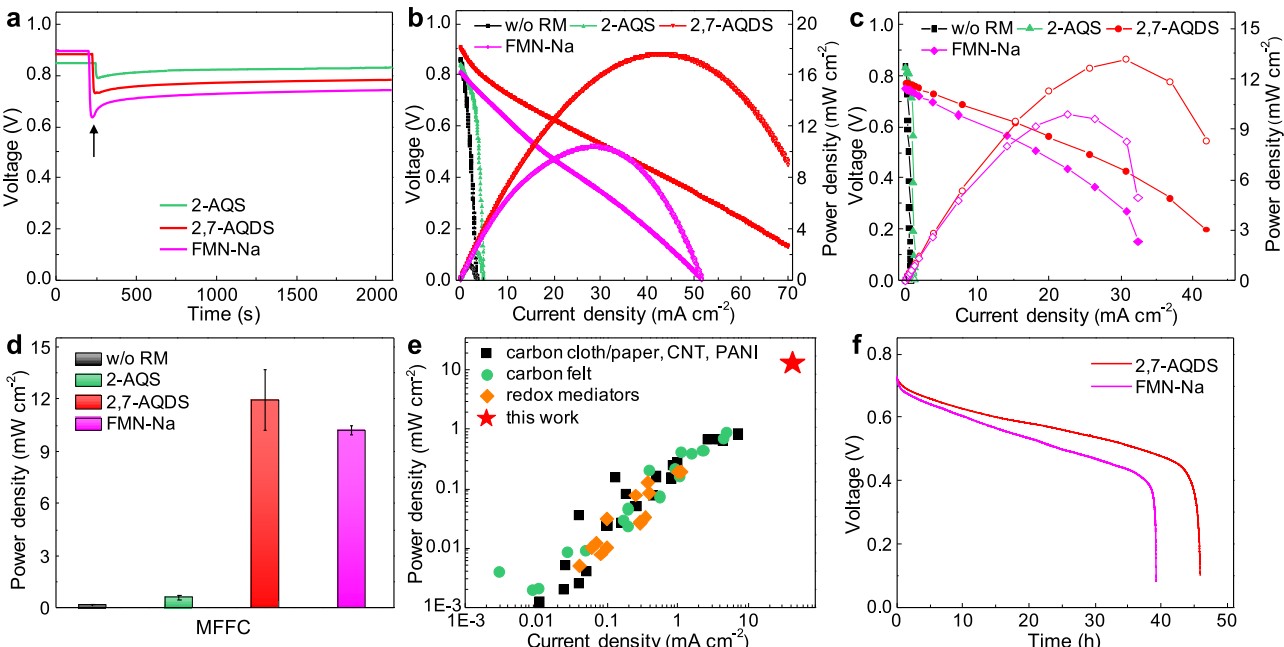

**Fig. 4 | Performance comparison of the MFFCs with or without redox mediators. a** OCVs of the MFFCs with different redox mediators (2-AQS: -0.1 mM, 2,7-AQDS: -25 mM, and FMN-Na: -25 mM). **b** Instant power polarization curves of the MFFCs with different redox mediators at the scan rate of 0.5 mA cm$^{-2}$ s$^{-1}$. **c** Polarization curves and power output of the redox-mediated MFFCs measured with external resistors. **d** Comparison of power densities of the MFFCs with or without redox mediators. The error bars represent the standard deviations from 3 repeating tests. **e** Comparison of power density and current density with the state-of-art MFCs with different electrode materials and redox mediators, respectively. **f** Long duration operation of the redox-mediated MFFCs under a constant current.

An ideal redox mediator should exhibit a reasonable solubility, appropriate redox potential, and satisfactory biocompatibility. With these considerations in mind and existing knowledges on the redox-active molecules[52], we initially identified three flavin- and quinone-based molecules as potential redox mediators, including riboflavin 5'-monophosphate sodium (FMN-Na), anthraquinone-2,7-disulfonic acid disodium (2,7-AQDS), and anthraquinone-2-disulfonic acid disodium (2-AQS), to explore the mediator effect on the bio-electrochemical reaction in the MFFC system (Fig. 3d). Considering some forms of flavins and quinones are naturally involved in the electron transfer chain of *S. oneidensis*, we expect these identified mediators may not undermine the bacterial viability and maintain redox activity within the bacteria medium[24].

We conducted cyclic voltammetry (CV) to evaluate the electrochemical properties of these redox mediators in aqueous solutions at neutral pH and compared against ferricyanide catholyte (Fig. 3e). The CV studies reveal that these selected redox species show a relatively symmetric CV curve, indicating reversible reactions. The redox mediators all exhibit a redox potential (E) of around -0.3 V (vs. SHE): [E$_{FMN-Na}$ (-0.15 V, -0.30 V) > E$_{2,7-AQDS}$ (-0.26 V) > E$_{2-AQS}$ (-0.33 V)], which is comparable to that of the lactate/CO$_2$ reaction in the bacteria metabolic oxidation process. Thus, the electrons can be transferred to redox mediators, thermodynamically. Further, we note that redox reactions of such redox mediators generally involve proton coupled electron transfer, and thus their redox potentials show a positive shift with decreasing pH based on Nernst equation[53]. Such a positive shift can boost electron transfer efficiency from bacteria to redox mediators, but may compromise the overall voltage output. Compared with the ferricyanide catholyte with a redox potential of -0.48 V, we could expect a maximum output potential of -0.8 V for the MFFCs constructed using these redox mediators. We have also evaluated the solubility of redox mediators in the bacteria medium (Fig. 3f), among which 2-AQS has a much lower solubility (< 0.1 mM) than those of 2,7-AQDS (~ 800 mM) and FMN-Na (-30 mM).

## The redox mediated MFFCs

To understand the interactions among the flowing bacteria, the redox mediators, and the electrodes, we have evaluated the OCV of the MFFCs with different redox mediators (Fig. 4a). Before introducing redox mediators, the MFFCs, generally show an OCV around 0.85 V, which drops instantly to 0.63–0.78 V when redox-active molecules are introduced, and the voltage drop region is about 100 sec before it reaches a stable OCV. The voltage decrease is mainly due to an extra step of electron transfer from bacteria to the redox mediators before reaching the carbon electrodes. As the redox mediator has the higher redox potential than *S. oneidensis* bacteria and the cathode potential is constant, the overall voltage decreases when the anode potential is determined by the redox mediator. This slight drop of OCV in redox mediated MFFCs is consistent with the expected voltage loss from the additional redox process. We note the OCV of MFFCs with different redox mediators shows a consistent trend (OCV$_{2-AQS}$ > OCV$_{2,7-AQDS}$ > OCV$_{FMN-Na}$) with the respective redox potentials (E$_{2-AQS}$ < E$_{2,7-AQDS}$ < E$_{FMN-Na}$): the more negative the redox potential, the larger the OCV. We note there is a transient increase of the OCV in the first few minutes of introducing the oxidized form of redox mediators (see arrows in Fig. 4a), which may be mainly attributed to the accumulative reduction of redox mediators before reaching an equilibrium concentration (Supplementary Figs. 6 and 7).

We next collected the instant polarization curve under a current scan rate of 0.5 mA cm$^{-2}$ s$^{-1}$ to evaluate the maximum power output of the redox mediated MFFCs (Fig. 4b). The polarization curves show nearly straight lines with a specific slope, suggesting a constant internal resistance and little concentration polarization at high current. Notably, with the redox mediators, the output current and power of the MFFCs are significantly improved, which can be attributed to the much more efficient electron extraction process. In particular, the MFFCs with 2,7-AQDS or FMN-Na showed a highest output current density beyond 50 mA cm$^{-2}$ and output power density of 17.6 or 10.4 mW cm$^{-2}$, respectively, which represents a 1275% or 712% increase

over those of the MFFC without redox mediators. The higher output power density achieved in the MFFC with 2,7-AQDS can be attributed to its lower redox potential, which leads to higher output potential.

The steady-state polarization curve obtained by loading a series of external resistors show consistent results (Fig. 4c, d). The power output trend among the MFFCs with different redox mediators correlates with solubilities and redox potentials of mediators. Importantly, the MFFC with 2,7-AQDS shows a maximum current output of ~42 mA cm$^{-2}$, which is considerably higher than that of other mediators and much higher than the MFFCs without redox mediators (<1 mA cm$^{-2}$). The maximum steady-state power density of the MFFCs with 2,7-AQDS or FMN-Na can reach ~13.1 or 10.2 mW cm$^{-2}$, about 50 times of that without redox mediators (~0.2 mW cm$^{-2}$). Importantly, the output current and power density achieved in the redox mediated MFFCs far exceed that of the current state-of-art MFC systems by one order of magnitude (Fig. 4e and Supplementary Table 1 and 2). We also found that a sufficient solubility of selected redox mediators (>5 mM) in the medium solution is necessary for boosting the charge extraction and transport efficiency and realizing high-power output (Supplementary Fig. 8). We note that the redox mediator strategy has been explored previously in conventional MFCs, although the achieved power output remains limited (< 0.4 mW cm$^{-2}$)[46,54–56] due to the solubility or mass transport limitations. We also measured the steady-state power output of MFCs with FMN-Na under the similar concentration (Supplementary Fig. 9), although the achieved power density (~0.49 mW cm$^{-2}$) is far lower than that of redox mediated MFFCs (>10 mWcm$^{-2}$), further confirming the merits of the MFFC design. Additionally, the control experiments on the MFFCs without bacteria but with redox mediators show no noticeable power output, confirming the power source is originated from the *S. oneidensis* metabolism (Supplementary Fig. 10).

The greatly enhanced current and power output highlight the critical role of redox mediators in accelerating the electron extraction process from the bacteria. For the FMN-Na, it is interesting that the polarization curves measured with resistor loading indicates the steady state output power (10.2 mW cm$^{-2}$) is only slightly lower than that of instant dynamic measurement with continuous current scan (10.4 mW cm$^{-2}$), indicating significantly reduced concentration polarization due to much increased redox molecule concentration (electron donors) to supply sufficient electrons for stable constant power output. This is in stark contrast to the MFFCs without additional redox mediators, where serious concentration polarization occurs (see Fig. 2c, d, 0.19 mW cm$^{-2}$ instant power vs. 1.28 mW cm$^{-2}$ steady state power). Similarly, the MFFCs with 2,7-AQDS showed a relatively small difference between the instant measurement (17.6 mW cm$^{-2}$) and the steady statement measurement (13.1 mW cm$^{-2}$), while the MFFCs with 2-AQS showed a more notable difference (2.18 mW cm$^{-2}$ instant power vs. 0.58 mW cm$^{-2}$ steady state power) (Supplementary Fig. 11) since its low solubility can lead to more notable concentration polarization.

We further conducted galvanostatic test to evaluate the stability of the redox mediator in the MFFC system under a constant output current by using an anion exchange membrane (AEM) (Fig. 4f). At the current density of 2.5 or 3 mA cm$^{-2}$, respectively, the MFFCs with the 2,7-AQDS or FMN-Na show a stable operation for a long period of ~40 h. The voltage output shows a gradual decrease during the operation period, which can be attributed to the cathode reduction, pH effect, and/or internal resistance increase during the galvanostatic discharge process. Based on the total charge output from the substrate consumed, we can calculate a Coulombic efficiency of 82.4 (2,7-AQDS) and 84.0% (FMN-Na) for the redox mediated MFFCs based on the four-electron path for the lactate substrate (as confirmed by NMR studies, Supplementary Fig. 12). Additionally, considering the total electrons extracted and total concentration of the redox mediators in the anolyte, we can determine a turnover number of about 50 for the redox

mediators, demonstrating highly reversible nature of the redox mediators in the MFFCs. Additional spectroscopy studies further confirm the long-term stability of the artificial redox mediators in the bacteria medium (Supplementary Fig. 13), and the antimicrobial activity of FMN-Na is largely negligible under our operation conditions (Supplementary Fig. 14). We note the ion exchange membranes can greatly affect the long-term operation of the redox-mediated MFFCs (Supplementary Fig. 15). With the proton exchange membrane (PEM) (i.e., Nafion), the mediated MFFCs can only sustain the operation for ~9 h at a constant current of 2.5 mA cm$^{-2}$, with a low Coulombic efficiency (16% on total lactate) and turnover number (9.6), which can be attributed to the excessive proton accumulation in the anode (pH = 5.2 after test).

## Internal resistance analysis

To further understand the role of redox mediators in the charge-transfer process, we have conducted the electrochemical impedance spectroscopy (EIS) to analyze the internal resistance of the MFFC in comparison with the MFC. As described in the equivalent circuit model (Supplementary Fig. 16), the internal resistance mainly includes three parts: ohmic (contact/solution/membrane), charge-transfer (cathode/anode reaction), and diffusion resistances. As the bio-electrochemical reaction of *S. oneidensis* is mainly dominated by the charge-transfer process, the Warburg diffusion element is not recognized in the Nyquist plots of the MFC or the MFFC systems with bare bacteria anolyte, while it is present in the system with redox mediators, because the reaction is affected by both the charge-transfer and diffusion processes[57–60] (Fig. 5 and Supplementary Figs. 17–19 and Table 3).

Without redox mediators, both the MFC and the MFFC show two semicircles in the EIS curves (Fig. 5a, b), which can be attributed to the charge-transfer reaction of the ferricyanide cathode and the bacteria anode, respectively. With the addition of redox mediators in the bacteria anolyte in MFFC systems, the EIS test shows a totally different Nyquist plot (Fig. 5c) with three parts: cathode reaction, anode reaction, and mass transport resistances. According to the fitting result, the first semicircle for all three plots, corresponding to the charge transfer of cathode, shows a comparable low resistance in the range of ~0.2-0.5 ohms, suggesting the overall internal resistance of the MFC or MFFC systems is mainly dominated by the anode reaction, as also identified by the EIS studies of separate cathode and anode (Supplementary Figs. 17 and 18).

The second semicircle represents the charge transfer process between bacteria and anode. Compared with the resistance of the biofilm in the MFC (~978 ohms) (Fig. 5a), the smaller resistance value (~281 ohms) observed in the MFFC (Fig. 5b) suggests that the flowing bacteria is more efficient in electron transfer process. More importantly, when the redox mediators are included, the anode charge-transfer resistance is greatly reduced to ~0.7 ohm (Fig. 5c), 2–3 orders of magnitude smaller than that observed in the MFC or the MFFC without redox mediators (Fig. 5d). This analysis confirms that redox mediators can vastly boost the extracellular electron transfer process and minimize the internal resistance, which is essential for achieving high output power. With the greatly reduced charge transfer resistance in the redox mediated MFFC, the mass transport resistance ($R_{MT}$ ~ 0.7 ohm) (Fig. 5c) starts to make a significant portion of the total resistance, which is not appreciable in the MFC or the MFFC without redox mediators due to the dominance by much larger charge transfer resistance. It is also confirmed that the determined internal resistances are generally consistent with the external resistances used at peak power densities (MFC, 995 ohms; MFFC, 330 ohms; mediated MFFC, ~3 ohms).

## Discussion

Together, we have reported a unique design of redox mediated MFFCs that allows efficient nutrient delivery to and electron extraction from

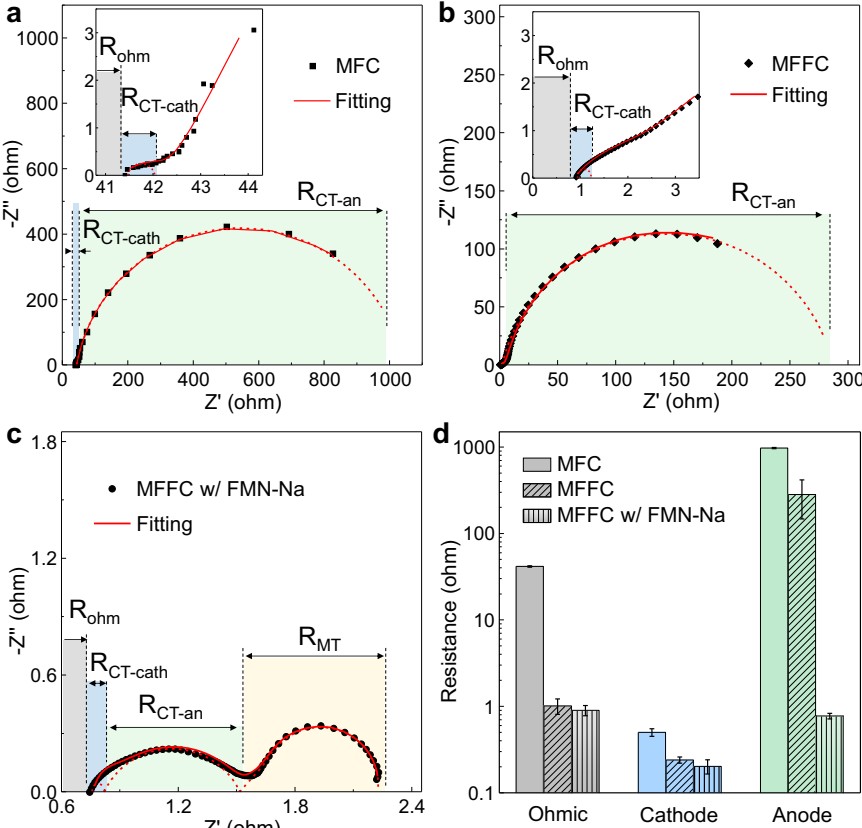

**Fig. 5 | Internal resistance analysis of MFCs and MFFCs with or without redox mediators. a** EIS analysis of the double-chamber MFC with carbon felt anode. **b** EIS analysis of the MFFC without redox mediators. The insets are the zoom-in view. **c** EIS analysis of the MFFC with FMN-Na as the redox mediator. **d** Resistance comparisons among the MFC and the MFFCs with or without FMN-Na (error bars represent standard deviations). The internal resistance includes the Ohmic, cathode reaction ($R_{CT-cath}$), anode reaction ($R_{CT-an}$), or mass transport ($R_{MT}$) resistances.

individual bacteria, achieving significant power output without the need of typical long-duration incubation of thick biofilms. In comparison with conventional MFCs that are often plagued with large charge-transfer resistance and nutrient transport limitations in thick biofilms, the redox mediated MFFCs exploit artificial redox mediators to facilitate more efficient electron transport, resulting in a greatly reduced internal resistance and one order of magnitude higher power output. The redox mediators can efficiently extract and store the metabolic electrons from the planktonic bacteria, and then transfer the electrons to the carbon felt electrodes, which breaks the intrinsic tradeoff between the requirement of thick biofilms for sufficient metabolic electron generation rate and mass/ion/electron transport limitation in dense biofilms. Thus, the redox mediated MFFC design could provide an effective pathway to break the fundamental limitations in traditional MFCs, and open an avenue for diverse bioelectrochemical technologies including microbial fuel cells, microbial electrolysis and biophotovoltaics. However, for practical applications, advanced cathode design is necessary in order to overcome the limited solubility of oxygen in aqueous solutions and boost cathode oxygen reduction reaction to match current output from the anode. Further, we have also evaluated the performance of redox-mediated MFFCs using the synthetic wastewater[61] and achieved a similar power output and stable operation (Supplementary Fig. 20). For the implementation in real wastewater, insoluble solids, limited organic concentration, diverse organic types, and low salinity in most real wastewater could compromise the system performance, which requires additional system engineering and optimization. Furthermore, the potential bacterial fouling on the anode carbon felt or the membrane could deteriorate mass transport and power output over time. It is also important to closely monitor the potential biofilm formation on carbon felt electrodes, which may block pore space to harm the interaction between redox mediators and the electrode and reduce the charge transfer efficiency. The development and adoption of proper antifouling strategies could be beneficial for long term operation.

## Methods

### Medium ingredient and bacteria culture

The *S. oneidensis* bacteria suspension with lactate as the nutrient is used as the model anolyte stored in an external reservoir, circulating between the reservoir and a porous carbon felt anode. *S. oneidensis* is easy to handle and can be robustly cultured in both aerobic and anaerobic environments, and both direct contact and indirect pathway mediated by redox mediators play an important role in the EET process. As a Gram-negative species with the electrically non-conductive outer membrane, the S. oneidensis can effectively highlight the beneficial role of flowing bacteria and artificial redox mediators in the MFFC design.

In general, the medium ingredient and bacteria culture process are similar to our previous report[20]. The *Shewanella* medium contains (per liter of deionized water): 9.07 g PIPES buffer ($C_8H_{18}N_2O_6S_2$), 3.4 g sodium hydroxide (NaOH), 1.5 g ammonium chloride (NH$_4$Cl), 0.1 g potassium chloride (KCl), 0.6 g sodium phosphate monobasic monohydrate (NaH$_2$PO$_4$·H$_2$O), 45 mM sodium DL-lactate (C$_3$H$_6$O$_3$) as the electron donor. This base medium is finally adjusted to a pH of 7.2 using HCl and NaOH and then sterile filtered using 0.22 μm vacuum driven disposable bottle top filters (Millipore). The components of synthetic wastewater[61] are as follows: 1120.6 mg L$^{-1}$ NaC$_3$H$_5$O$_3$, 191.1 mg L$^{-1}$ NH$_4$Cl, 30.8 mg L$^{-1}$ KH$_2$PO$_4$, 19.3 mg L$^{-1}$ CaCl$_2$·2H$_2$O,

71 mg L$^{-1}$ MgSO$_4$·7H$_2$O, 17.4 mg L$^{-1}$ FeSO$_4$·7H$_2$O, 0.025 mg L$^{-1}$ H$_3$BO$_3$, 0.07 mg L$^{-1}$ CuCl$_2$·2H$_2$O, 0.033 mg L$^{-1}$ KI, 0.013 mg L$^{-1}$ MnCl$_2$·4H$_2$O, 0.016 mg L$^{-1}$ Na$_2$MoO$_4$·2H$_2$O, 0.013 mg L$^{-1}$ ZnSO$_4$·7H$_2$O, 0.03 mg L$^{-1}$ CoCl$_2$·6H$_2$O.

*Shewanella oneidensis* MR-1 is first inoculated in the Luria Broth (LB) solution (20 ~ 30 ml). The flask with the LB solution is placed in a 30 °C shaker for 12 ~ 20 h (usually overnight). Before any test, ~10 ml of bacteria colonies is taken out, centrifuged (1378 × *g*, 5–10 mins) and washed 2–3 times with *Shewanella* medium. For the electrochemical test, the bacteria colonies are added into ~40 ml medium. The final bacteria solution's OD$_{600\ nm}$ is about 1.0 ± 0.2.

### Cyclic voltammetry (CV) measurement

The CV measurements of redox mediators and potassium ferricyanide were conducted on the BioLogic VMP3 potentiostat system at room temperature, using the three-electrode configuration with glassy carbon as working electrode (diameter 3 mm diameter), Ag/AgCl (3 M KCl) as reference electrode and Pt wire as counter electrode, respectively. The Ag/AgCl reference electrode potential is ~0.21 V vs. SHE. According to the CV curves, the half-wave potentials are recorded as redox potentials of redox mediators. The scan rate for all CV tests is 10 mV s$^{-1}$. Generally, for the CV test at pH = 7, the used electrolyte is 5 mM redox mediators or potassium ferricyanide in 1 M NaCl aqueous solutions, while for the Anthraquinone-2-disulfonic acid disodium (2-AQS) mediator, the used electrolyte is saturated 2-AQS (adding 0.5 mM) in 0.5 M NaCl aqueous solutions. For the CV test at low pH, the used electrolyte is 1 mM redox mediators in 1 M NaCl aqueous solutions with adjusting pH values by adding H$_2$SO$_4$.

### Microbial flow fuel cell (MFFC) experiment

The MFFC experiments were carried out using the flow battery device comprised of poly(tetrafluoroethylene) (PTFE) frame, graphite plates current collector, and carbon felt electrodes with an active area of 2 × 2 cm$^2$. The anode and cathode areas are the same as the area of the membrane channel. The Nafion 211 or 212 membranes are used to separate the cathode and anode with conducting cations. The membrane was pretreated in NaOH solutions at 80 °C for 2 h. In the MFFC test, the bacteria solution (~40 ml) with lactate as electron donor is used in the anode side, named as anolyte, and the potassium ferricyanide solution with sodium chloride is used in the cathode, named as catholyte (~40 ml). For the MFFC test without redox mediators, aqueous solution of 0.1 M K$_3$Fe(CN)$_6$ and 0.2 M NaCl was used as the catholyte. For the MFFC test with redox mediators, an aqueous solution of 0.3 M K$_3$Fe(CN)$_6$ and 0.5/1.0 M NaCl was used as the catholyte. In the power measurement, the applied concentration of redox mediators is ~20–25 mM for Riboflavin 5'-monophosphate sodium (FMN-Na) and Anthraquinone-2,7-disulfonic acid disodium (2,7-AQDS), while for the 2-AQS mediator, the used concentration is about 0.1 mM due to its solubility limit. For all the MFFC experiments, the anolyte and catholyte are circulated at a flow rate of ~20 ml min$^{-1}$. All the electrochemical tests are conducted by the CHI electrochemical workstation. The open circuit voltage is directly recorded without an external resistor. The polarization curves are obtained by varying external resistors or changing the applied current at the scan rate of 0.5 mA cm$^{-2}$ s$^{-1}$. The output voltage is recorded by connecting external resistors. The power calculation is based on the projected area. All MFC experiments are operated at room temperature (20 ~ 25 °C).

For the long-standing test to determine the Coulombic efficiency and turnover number of redox mediators, the experiment was conducted in a glove box with N$_2$ gas protection, and the Nafion or anion exchange membrane (AEM) (Fumasep FAS-30) were used in the MFFC system. Note that the lactate concentration was increased to 60 mM in the medium solution for the long-term test. The applied anolyte and catholyte are 120 ml bacteria solution with 3 mM FMN-Na or 2,7-AQDS and 120 ml 0.3 M K$_3$Fe(CN)$_6$ solution with 0.5 ~ 1 M NaOH, respectively.

This long-standing experiment is based on the galvanostatic measurement with applying a constant current (10 or 12 mA). During the measurement, the applied flow rate is ~20 ml min$^{-1}$.

### Conventional microbial fuel cell (MFC) experiment

The H-type MFC is constructed by connecting two chambers (~120 ml) with a proton exchange membrane (PEM) separator (Nafion 211 or 212). The diameter of the chamber channel is 3.2 cm. The current collector in the cathode side is carbon cloth (2 cm × 7 ~ 8 cm). The carbon felt (CF) is cut into small pieces, 4 cm$^2$ (2 cm × 2 cm), used as the anode current collector. Before the tests, the sterilization process is conducted. Before constructing the devices, the chambers and other components are soaked in aqueous detergent solution and rinsed with deionized water sufficiently. The applied anolyte is ~100 ml bacteria solution. All MFC experiments are also operated at room temperature (~20–25 °C). At the steady state, the polarization curves are obtained by varying the external resistor or changing the applied current at the scan rate of 0.5 mA cm$^{-2}$ s$^{-1}$. In the measurement with external resistors, the output current is calculated based on Ohm's Law: I = V/R (R is the value of external resistors). The output power is calculated by P = IV. In all tests, the cathode solution is potassium ferricyanide (K$_3$Fe(CN)$_6$, 0.1 ~ 0.3 M) and sodium chloride (NaCl, ~0.2–1 M).

### Electrochemical impedance spectroscopy (EIS) measurement

To analyze the resistance of full MFC and MFFC, the EIS measurement is generally conducted after the performance test. The catholyte is the potassium ferricyanide with sodium chloride, and the anolyte is the bacteria solution with or without redox mediators. In the EIS test, the cathode is connected with the working electrode of CHI electrochemical workstation. For the EIS measurement based on the three-electrode setup, the measurement is conducted with the three-electrode configuration, using carbon felt or carbon cloth as the working electrode. The Pt wire and Ag/AgCl are used as the counter electrode and the reference electrode, respectively. All three electrodes are immersed in the same electrolyte. In general, EIS is measured with the frequency range of 10$^5$–0.01 Hz at the open circuit voltage.

### Antimicrobial activity of FMN-Na

We investigated the antimicrobial activity of the FMN-Na mediator against *S. oneidensis* by culturing the bacteria in the LB-Agar plate or lactate medium with different concentrations of the redox mediator. For the LB-Agar plate, the applied concentrations are 0, 5, 15, 25, 50, 75 mM. For the lactate medium, the applied concentrations are 0, 3, 5, 15, 25, 35 mM. The bacteria are first cultured from the original plate in the fresh LB solution for ~17 h, and are then inoculated into the LB-Agar plate or lactate medium with different concentrations of FMN-Na. For the incubation on the LB-Agar plate, it was placed in a 30 °C oven, and the bacteria growth was observed visually to determine if the FMN-Na has the apparent antimicrobial activity. In the lactate medium, the bacteria were cultured for a period of time and then centrifugated (2500 × *g*, 10 mins) to remove the redox mediator and redispersed into the same amount of fresh medium for the OD test.

### Scanning electron microscope (SEM) and confocal laser scanning microscopy (CLSM) characterizations

The microstructure of carbon felt electrode was directly characterized via ZEISS Supra 40VP SEM. To study the interaction of bacteria and carbon felt electrode during the continuous flowing of bacteria solution, the entire electrode after flowing for a specific time (i.e., 10 mins, 6 h, 24 h) is fixed by 2.5% glutaraldehyde (C$_5$H$_8$O$_2$) solution overnight and chemically dehydrated using gradient concentration ethanol aqueous solutions (50, 70, 80, 90, and 95% each one time, then 100% twice). The sample was finally dried in the air and then randomly cut into thin pieces for further characterizing. The gold sputter-coating (15 mA, 30 s) was also conducted before SEM for the better imaging.

To further characterize the viability of bacteria that interacted with carbon felt electrode during the continuous flowing, the L7007 live/dead bacterial viability kit is utilized. After flowing for a period of time, a piece of electrode is rinsed with PBS and stained with SYTO 9 dye and propidium iodide (PI) mix solution. Finally, the electrode with stained cells is characterized by the Leica TCS SP8 II confocal/multi-photon laser scanning microscope (Germany). The live-dead assay solution is obtained by diluting SYTO 9 dye solution (1.67 mM solution in dimethyl sulfoxide) and propidium iodide (1.67 mM solution in dimethyl sulfoxide) stock solutions in PBS at final concentrations of 30 µM and 30 µM. For the living bacteria, they will show strong green fluorescence in the image.

## Lactate and acetate concentration determination

The lactate and acetate concentrations after the electrochemical test are determined by the nuclear magnetic resonance (NMR) analysis. $D_2O$ is used as deuterated solvent and TMSP is used as the internal standard. Generally, a 1 mM TMSP/$D_2O$ solution is prepared before the sample preparation. For the NMR test, 0.6 ml target solution is mixed well with 0.1 ml 1 mM TMSP/$D_2O$ solution. The target solution is obtained by centrifuging the bacteria anolyte for 30 mins at ~4365–9821 × $g$ for three times with removing bacteria completely and stored in the fridge (~4 °C).

## Turnover number (TON) and Coulombic efficiency (CE) calculations

The Coulombic efficiency is evaluated by the following equations. In Eq. (1), the η is the Coulombic efficiency; $C_{output}$ is the amount of experimental electric quantity, in Coulombs; $C_{theo}$ is the theoretical amount of charge (Q, in Coulombs) based on the consumed amount of lactate that determined by NMR spectroscopy. As the final product is acetate, the calculation is based on four electrons transferred. In Eq. (2), $A_{lactate}$ is the lactate concentration consumed in the bacteria medium during the test. $V$ is the total volume of the medium. $N$ is the number of electrons transferred during the oxidation of one molecule of lactate. $F$ is Faraday's constant (96500 C mol$^{-1}$).

$$\eta = C_{output}/C_{theo} \tag{1}$$

$$C_{theo} = A_{lactate} \times V \times N \times F \tag{2}$$

The turnover number of single redox mediator is evaluated using Eq. (3). Here, $n_{substrate}$ is the total number of electrons experimentally delivered from the lactate oxidization; $n_{catalyst}$ is the number of catalysts (i.e., the mediator number).

$$Turnover\ number = n_{substrate}/n_{catalyst} \tag{3}$$

## Data availability

All data needed to evaluate the conclusions in the paper are present in the paper and/or the Supplementary Information. Source data are provided with this paper, https://doi.org/10.6084/m9.figshare.24796686.

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

## Acknowledgements

Y.H. acknowledges the support from Office of Naval Research (grant no. N00010141712608). We acknowledge the Advanced Light Microscopy/Spectroscopy at UCLA for CLSM technical support. The authors thank Ao Zhang for his help on SEM characterizations.

## Author contributions

Y.H. and X.D. conceived the research. L.Z., X.D., and Y.H. designed the experiments. L.Z. performed the experiments and conducted data analysis. Y.Z. cultured the *S. oneidensis* bacteria with the assistance of C.L. and collected the CLSM images. Y.L. collected the SEM images. S.W. conducted the NMR analysis. Y.H. and X.D. supervised the research. L.Z., X.D., and Y.H. co-wrote the paper with input from all authors.

## Competing interests

The authors declare no competing interests.
