## [Peer Review File · Nature Communications]

fuel cell system that is able to bypass current limitations with microbial fuel cell technologies. Namely, this article addresses the issues presented by biofilm formation in MFCs, such as high internal resistance, slow maturation rates and conductivity issues (from non-conductive debris, biofouling and catalyst inactivation). This article's approach, uses a flow system which removes the need of biofilm formation for electricity generation by exoelectrogenic bacteria. The technology utilised results in a one order of magnitude increase in power outputs of microbial fuel cell technologies. This article is timely, novel and provides evidence towards industrial realisation of Microbial fuel cell technologies. The manuscript is well written and will make an excellent addition to Nature Communications following some minor corrections (as detailed below).

Line 1 – The title is too broad and more details should be included (what bacterial species) what are the key findings etc.

Line 26 – the statement 'approximately one order of magnitude higher' should be countered with, 'to the best of the authors knowledge'.

Line 26 – To ensure that this MFFC configuration does indeed produce a power density that is an order of magnitude higher than current state of the art MFCs I would consider including a table of the highest performing MFCs to date – including the results from this study.

Line 31 – Gram should be capitalised here and throughout the manuscript.

Line 31 – Shewanella species (*S. oneidensis*) should be stated throughout the manuscript, simply stating Shewanella is not enough.

Line 40 – temperature is also a factor that effects bacterial tolerance and this should be stated.

Line 43-44 – a reference is required (H-shaped MFCs).

Line 49 - I would argue that delicate is not the correct word here, sensitive to environmental pressure yes, but not delicate.

Line 50 - A paragraph on biofilm conductivity issues should be included - this will provide further evidence as to why this design is advantageous to a more traditional biofilm focussed approach.
Literature of interest:

[1] Sun M, Zhai L-F, Li W-W, Yu H-Q. Harvest and utilization of chemical energy in wastes by microbial fuel cells. *Chem Soc Rev* 2016;45:2847–70.

[2] Blanchet E, Erable B, De Solan M-L, Bergel A. Two-dimensional carbon cloth and three-dimensional carbon felt perform similarly to form bioanode fed with food waste. *Electrochem Commun* 2016;66:38–41.

[3] Islam MA, Woon CW, Ethiraj B, Cheng CK, Yousuf A, Khan MMR. Ultrasound driven biofilm removal for

Line 507 – Please ensure the referencing style adheres to that of Nature Communications.

electrode as thick biofilms, in which electrons are transferred onto carbon electrode via direct contact or naturally secreted redox shuttle (e.g., flavins). In comparison, in the MFFC, *Shewanella* suspension is flowed through porous carbon felt electrodes, in which *Shewanella* bacteria have a dynamic collision interaction with the electrode to induce dynamic collision electron transfer rather than a nearly static interaction in biofilms. The additional redox mediators with a particle density ($1.8-15 \times 10^{18}/\text{cm}^3$) ~ 10 orders magnitude higher than bacteria concentration ($8 \times 10^8/\text{cm}^3$), can greatly boost the collision charge transfer efficiency from **bacteria->mediators** and from **mediators->carbon felt electrode**, leading to greatly increased current and power density.

7. The "Methods" section recommends providing the total volume of MFFC used, along with the quantity of anolyte and catholyte used.

Response: Thanks for the comment. We have provided more details in the method section with including the volume of anolyte and catholyte used.

Response: Thanks for the comment. We have removed this sentence.

Line 409 – how long were the membranes pre-treated at 80 °C for?

Response: Thanks for the comment. The membranes were pretreated at 80 °C for 2 hours. We have added this information in the method section.

Line 410 – was the catholyte also at a 40 mL volume?

Response: Thanks for the comment. The volume of used catholyte is kept the same as that of the analyte (i.e., 40 mL). We have added the volume number for the catholyte.

Line 411 – Sentence should be reworded as it is difficult to follow.

Response: Thanks, and we have revised.

Line 463 – parameters of gold sputter coating should be included (more details required (voltage, time.. etc.).

Response: Thanks for the comment. We have added the parameters of gold sputter coating in the method section (15 mA, 30 s).

Line 468 – replace 'alive' with live.

Response: Thanks for the comment. We have replaced it as suggested.

Line 479 – 'in the fridge' what was the temperature – 4°C?

Response: Thanks for the comment. The fridge temperature range is 3-5 °C. The working temperature is around 4 °C. We have added this temperature value in the sentence mentioned by the reviewer.

Line 507 – Please ensure the referencing style adheres to that of Nature Communications.

Response: Thanks for the comment. We have checked the referencing style to ensure consistency with *Nature Communications* style.

Reviewers' comments:

Reviewer #1 (Remarks to the Author):

attached

Reviewer #2 (Remarks to the Author):

I am satisfied with responses, I recommend it to the journal for further processing.

Reviewer #3 (Remarks to the Author):

This reviewer would like to thank the authors for addressing all concerns and comments in an elegant and comprehensive manner. I have no further suggestions and I would advise publication of this article without hesitation to Nature Communications.